# Sex Differences in Glomerular Lesions, in Atherosclerosis Progression, and in the Response to Angiotensin-Converting Enzyme Inhibitors in the ApoE^−/−^ Mice Model

**DOI:** 10.3390/ijms241713442

**Published:** 2023-08-30

**Authors:** Adrián Mallén, Ronny Rodriguez-Urquia, Rafael Alvarez, Eduard Dorca-Duch, Estanis Navarro, Miguel Hueso

**Affiliations:** 1Experimental Nephrology Laboratory, Institut d’Investigació Biomèdica de Bellvitge-IDIBELL, 08907 L’Hospitalet de Llobregat, Spain; amallen@idibell.cat (A.M.); estanis.navarro@gmail.com (E.N.); 2Department of Nephrology, Hospital Universitari Bellvitge, 08907 L’Hospitalet de Llobregat, Spain; rrogriguez@hospitalbellvitge.cat (R.R.-U.); ralvareze@bellvitgehospital.cat (R.A.); 3Department of Pathology, Hospital Universitari Bellvitge, 08907 L’Hospitalet de Llobregat, Spain; edorca@bellvitgehospital.cat

**Keywords:** sex differences, chronic kidney disease, atherosclerosis, foam cells, parietal epithelial cells (PECs), Angiotensin-Converting Enzyme inhibitors (ACEi)

## Abstract

This study analyzes sex-based differences in renal structure and the response to the Angiotensin-Converting Enzyme (ACE) inhibitor enalapril in a mouse model of atherosclerosis. Eight weeks old ApoE^−/−^ mice received enalapril (5 mg/kg/day, subcutaneous) or PBS (control) for an additional 14 weeks. Each group consisted of six males and six females. Females exhibited elevated LDL-cholesterol levels, while males presented higher creatinine levels and proteinuria. Enalapril effectively reduced blood pressure in both groups, but proteinuria decreased significantly only in females. Plaque size analysis and assessment of kidney inflammation revealed no significant sex-based differences. However, males displayed more severe glomerular injury, with increased mesangial expansion, mesangiolysis, glomerular foam cells, and activated parietal epithelial cells (PECs). Enalapril mitigated mesangial expansion, glomerular inflammation (particularly in the female group), and hypertrophy of the PECs in males. This study demonstrates sex-based differences in the response to enalapril in a mouse model of atherosclerosis. Males exhibited more severe glomerular injury, while enalapril provided renal protection, particularly in females. These findings suggest potential sex-specific considerations for ACE inhibitor therapy in chronic kidney disease and atherosclerosis cardiovascular disease. Further research is needed to elucidate the underlying mechanism behind these observations.

## 1. Introduction

Atherosclerosis (ATH) is the leading cause of vascular disease [1], and a growing body of evidence suggests that a decline in renal function plays a significant role in the development of ATH [2]. Furthermore, systemic hypertension, often associated with renal diseases, not only increases the risk of renal injury [3] but is also considered a modifiable risk factor for ATH [1]. One key mechanism implicated in hypertension, renal injury, and ATH is angiotensin II (Ang II), the principal effector of the Renin-Angiotensin system (RAS) [4]. Ang II contributes to fibrosis and inflammation, further exacerbating the damage [5], and RAS blockage has been shown to decrease the formation and progression of ATH [6] while providing renoprotection [4]. In this sense, in a mice model of ATH and reduced renal mass, the administration of losartan (an AT1-R blocker) between weeks 12 and 24, reduced the development of ATH, macrophage migration, and intimal infiltration [7].

It is known that women have a lower risk for cardiovascular or chronic kidney diseases (CKD) and show a slower progression to end-stage renal diseases (ESRD) after an injury than men [8,9]. Furthermore, it has been suggested that differences in responsiveness to Ang II [10] are responsible. Thus, in several cardiovascular studies, women appeared to be less responsive to Angiotensin-Converting Enzyme inhibition (ACEi) than men [11,12], while women with proteinuria responded better to treatment with ACEi than men [13]. Lastly, men required larger doses of AT1-R blocker to achieve the same blood-pressure-lowering effect as women [14].

New technologies such as the Whole Slide Imaging (WSI) that create high-resolution images, together with artificial intelligence (AI)-based slide analysis have contributed to the emergence of digital pathology, which allows a reduction in interobserver and intra-observer variability [15]. In renal pathology, AI-based automated high-throughput methods using WSIs have demonstrated advantages over traditional manual reads to study the glomerulus [16,17] or for the quantification of fibrosis lesions [18].

The ApoE^−/−^ mouse is a standard model to study ATH and has also been proposed as a model of renal injury [19]. In this work, we aimed to evaluate sex-associated differences in the progression of atherosclerosis, renal dysfunction, and the response to treatment with inhibitors of RAS in ApoE^−/−^ mice, using automated computer-assisted image analysis to obtain more precise and reproducible results.

## 2. Results

In the present work, mice fed with a western-style diet were treated with the RAS inhibitor, enalapril, and several metabolic determinations and analyses were performed in males and in females with the aim to detect sex-based differences in ATH progression or treatment-related regression.

### 2.1. Blood Lipids Profile in the ApoE^−/−^ Mice Model

At the beginning of the experiment, untreated female mice presented significantly increased low-density lipoprotein (LDL)-cholesterol levels when compared with males (25.96 ± 8.11 mmol/L in females vs. 11.45 ± 3.96 mmol/L in males, *p* = 0.02) but similar high-density lipoprotein (HDL)-cholesterol (2.28 ± 0.4 mmol/L in females vs. 2.38 ± 0.4 mmol/L in males, *p* = ns) or triglycerides (TG) levels (52.28 ± 15.89 mg/dL in females vs. 84.20 ± 21.49 mg/dL in males, *p* = 0.5). After treatment with enalapril for 4 weeks, females still presented significantly increased LDL-cholesterol (19.92 ± 10.98 mmol/L in females vs. 11.68 ± 5.17 mmol/L in males, *p* = 0.008) or TG levels (59.8 ± 3 mg/dL in females vs. 57.7 ± 7.98 mg/dL in males, *p* = 0.045) but similar HDL-cholesterol levels (2.44 ± 0.13 mmol/L in females vs. 2.64 ± 0.14 mmol/L in males, *p* = ns); see also Table 1.

### 2.2. Sex-Related Differences in Blood Pressure and Renal Function

Male mice displayed significantly higher creatinine levels (4.06 ± 2.1 mg/dL in the male group vs. 1.1 ± 1.09 mg/dL in the female group, *p* = 0.012) and proteinuria (1122 ± 498 mg/dL in the male group vs. 122 ± 58 mg/dL in the female group, *p* = 0.016) when compared with the female group, while systolic blood pressure (BP) did not show significant variations with regard to sex (137 ± 12 mmHg in female group vs. 107 ± 25 mmHg in male group, *p* = 0.2). After treatment with enalapril for 4 weeks, BP was found to have lowered in both groups (108 ± 12 mmHg in the female group vs. 101 ± 15 mmHg in the male group; *p* = 0.7). On the other hand, creatinine showed a different evolution in both groups because its values increased in females but decreased in males (2.04 ± 2.53 mg/dL in the female group vs. 1.72 ± 1.46 mg/dL in the male group, *p* = 0.5), while proteinuria decreased in females but not in males (105 ± 78 mg/dL in the female group vs. 1146 ± 290 mg/dL in the male group; *p* = 0.003); see also Table 1.

### 2.3. Sex-Related Differences in Control and Enalapril-Treated Mice

We did not observe significant sex-related differences in plaque size of aortic origin among males and females quantified by cross-section with the Oil-Red-O (ORO) method and expressed as the sum of all the individual percent of surface areas measured throughout the valve (42.22 ± 7.62% in the female group vs. 31.34 ± 10.33% in the male group, *p* = 0.7). Figure 1A shows a representative histologic stain on an aortic root cross-section evidencing an ATH lesion. Figure 1B documents their quantification, and Figure 1C measures lesions in the entire aorta from both groups, measured as the percent of surface area after “en face” staining. In addition, Figure 1D,E show a similar content of collagen in the lesions, assessed by Masson trichrome staining (TC) in both groups (38.67 ± 8.99% in the female group vs. 36.34 ± 8.18% in the male group; *p* = 0.8). Furthermore, we observed a similar degree of plaque inflammation, as measured by the proportion of macrophages F4/80+ (19.6 ± 0.03% in the female group vs. 18.7 ± 0.04% in the male group, *p* = 0.4) or the NFκΒ+ cells (21.7 ± 0.05% in the female group vs. 23.3 ± 0.06% in the male group, *p* = 0.6).

After treatment with enalapril for 4 weeks, results were similar in the two groups for lesion sizes (42.68 ± 3.79% in the female group vs. 42.26 ± 9.28% in the male group, *p* = 0.9) or collagen content (36.16 ± 8.40% in the female group vs. 43.56 ± 8.04% in the male group, *p* = 0.9). On the other hand, the “en face” analysis of the entire aorta did not show significant variations among the PBS control group (7.03 ± 4.15% in the female group vs. 11.9 ± 4.01% in the male group; *p* = 0.27) or the enalapril-treated group (3.75 ± 4.75% in the female group vs. 4.11 ± 1.30% in the male group; *p* = 0.71). However, differences were detected in the enalapril treatment in the male group (11.9 ± 4.01% in the male control group vs. 4.11 ± 1.30 in the male enalapril-treated group; *p* = 0.0029). 

### 2.4. Male ApoE^−/−^ Mice Showed Increased Number of Glomerular Foam Cells and Hypertrophy of PECs, While Enalapril Treatment Reduced Mesangial Matrix Expansion and Inflammation

Since serum creatinine and proteinuria were significantly increased in the male mice, we looked for renal functional or structural alterations that could justify these results. Although we did not find any globally sclerotic glomerulus, light microscopy revealed a number of structural findings in the male mice compatible with a more severe renal injury, such as an increased glomerular area (Figure 2B) and a reduction in the urinary space in the control group (26.19 ± 0.91% in female group vs. 16.31 ± 4% in male group, *p* = 0.015) and the enalapril-treated group (32.64 ± 2.57% in female group vs. 17.45 ± 2.69% in male group, *p* < 0.0001), thickening of the glomerular basement membrane, and expansion of the mesangial matrix with mesangiolysis in conjunction with the accumulation of glomerular foam cells (Figure 2A). Nevertheless, measures of mean mesangial cellularity (Figure 2C) did not show significant differences among groups. In the morphological analysis, light microscopy revealed that segmental hypertrophy of parietal epithelial cells (PECs) was more prominent in males in both the PBS group (0.2 ± 0.13% in female group vs. 4.07 ± 1.64% in male group, *p* = 0.0022) and the enalapril-treated group (0.13 ± 0.05% in female group vs. 4.07 ± 1.64% in male group, *p* = 0.0048) but differences between the PBS and enalapril groups were not significant (Figure 2D).

On the other hand, and in order to evaluate the degree of inflammation in the kidney, we counted the number of glomerular infiltrating monocytes (F4/80+ cells) and the number of activated NF-κB+ cells. Females showed a nonsignificant increase in infiltrating monocytes (12.79 ± 8.91% in female group vs. 7.75 ± 5.02% in male group, *p* = 0.14) (Figure 2F), and the activation of glomerular NF-κB+ was also similar (29.84 ± 23.46% in female group vs. 29.45 ± 31.94% in male group, *p* = 0.57) as shown in Figure 2H. Tubulointerstitial damage (tubular atrophy and interstitial fibrosis) were similar (not shown). Furthermore, mice treated with ACEi enalapril showed a reduction in the mesangial matrix that increased in the urinary space, particularly in females (32.85 ± 2.57% in female group vs. 17.46 ± 2.69% in male group, *p* = 0.0005). Endocapilar and mesangial hypercellularity were similar in both groups (Figure 2C), suggesting a reduction in glomerular pressure. In the females, enalapril also reduced the number of F4/80 + cells when compared with the control group (12.79 ± 8.91% in controls vs. 10.08 ± 7.87% in treated group, *p* = 0.2) but not in males (7.75 ± 5.02% in controls vs. 7.36 ± 12.86% in treated group, *p* = 0.42). Activation of glomerular NF-κB in the enalapril-treated group was reduced compared to the PBS group in both female (29.84 ± 23.46% in female control group vs. 2.33 ± 1.47% in treated female group, *p* = 0.001) and male (29.45 ± 31.94% in control male group vs. 8.14 ± 6.86% in treated male group, *p* = 0.22), suggesting an improvement in the overall degree of inflammation. Mice treated with enalapril also reduce the hypertrophy of PECs in the male and female mice.

Lastly, we also investigated the concordance between the calculation of the glomerular area and urinary space with deep learning software based on the MIB2 image semantic segmentation software [20]. The trained and validated network DeepLabV3 with a ResNet18 backbone and 2D U-Net architecture using QuPath 349 × 349 × 3 .tif images and .png masks (Figure 3B) revealed a good prediction of classes (global accuracy of 0.746) as showed in Figure 3D. Automatic identification of the glomerular area and urinary space also showed a reduction in the urinary space in the male mice in both the PBS group (22.77 ± 1.53% in female group vs. 19.28 ± 4.75% in male group, *p* = 0.0003) and the enalapril-treated group (26.19 ± 2.51% in female group vs. 16.72 ± 1.44%, *p* < 0.0001) as showed in Figure 3E. The correlation graphs revealed that manual ImageJ2 v 2.9.0/1.53t and automatic DeepMIB2 quantification tools have a good correlation in both the PBS (R = 0.54) and enalapril-treated (R = 0.76) groups (Figure 3F).

## 3. Discussion

In this study, we focused on sex-associated differences in renal function and structure using a mouse model of hypercholesterolemia and ATH since a strong link between CKD and atherosclerotic cardiovascular diseases (ASCVD) has been proposed. In our experimental model we did not observe any significant differences in the degree of inflammation or plaque size between the male and female mice. Additionally, treatment with enalapril did not have a significant effect on plaque size measured with ORO in the aortic root. Notably, our main finding was the association between glomerular foam cell accumulation and podocyte injury, which was evident in the male but not the female mice. Furthermore, treatment with enalapril reversed PEC hypertrophy. 

ATH and hypertension are associated with alterations in the structure and function of the vascular wall [21]. In contrast, clinical and experimental studies have demonstrated that RAS inhibitors induce vascular remodeling and effectively hinder the progression of ATH [22,23,24]. Furthermore, experimental data indicated that high doses of candesartan, but not normal doses, allow for the regression of established ATH lesions in the ApoE^−/−^ mice. Importantly, this regression occurred without causing significant changes in systolic blood pressure, serum total cholesterol, or HDL-cholesterol levels [25]. However, the response to ACEi can differ among arteries, implying variations in the specific composition within and between vascular regions [21]. This variability in the ACEi response can also be influenced by factors such as previous vascular lesions, aging, sex, or the species used in the experimental model. In our experiment, we observed that treatment with enalapril for 4 weeks reduced lipid staining in the whole aorta of male animals but did not have a significant effect on plaque size in the aortic root. This lower extension of ATH in the entire aorta may reflect a less severe condition. While plaque formation primarily initiates in the aortic origin, where turbulent flow patterns and more pronounced pressure differences compared to the distal aorta, animals with extensive ATH also exhibited additional plaques in the abdominal aorta [26]. Additionally, previous studies have reported slower lesion development in these specific aortic segments [27]. Moreover, ApoE transfer experiments highlighted a decrease in lipid content mainly in these aortic segments [28]. 

Most preclinical studies using mice models to study ATH did not examine both sexes. Even when both sexes were considered, well-powered direct statistical comparisons for sex as an independent variable were infrequently conducted [8]. Despite these limitations, male animals are noted to develop more inflamed plaque that is, in contrast, smaller compared to female animals [8]. However, after 8 months of age in ApoE^−/−^ mice, several studies have observed that males exhibit equal or even larger plaques [8]. In our experiment, we did not observe any significant sex-related differences in plaque size measured in the aortic root in 22-week-old ApoE^−/−^ mice fed for 14 weeks with a high-fat diet (Figure 1B).

Some evidence suggests that components of the endogenous RAS may be different in males and females and that sex steroids can modulate the expression and activity of the various components of the RAS in the kidney and other tissues [29]. In addition, when a functional RAS is present, Ang II infusion in mice and rats resulted in greater increases in BP in normotensive males compared to females [30]. Furthermore, when endogenous RAS was blocked by enalapril in normotensive Sprague-Dawley rats, the infusion of Ang II increased albuminuria in males but not in females, suggesting that females were protected from renal injury when given Ang II and high salt. Conversely, the increase in renal injury in males contributed to the exacerbation of BP in the presence of Ang II and high salt [31]. Our results were in accordance with this data since male mice exhibited increased albuminuria, but we did not detect differences in BP when compared to females, suggesting the involvement of other factors. Notably, in our experiment, males exhibited lower cholesterol levels compared to females. Additionally, males treated with enalapril showed lower TG levels (Table 1).

Next, we focused on the kidney. It has been proposed that Focal Segmental Glomerulosclerosis (FSGS) was similar to ATH based on similarities of mesangial cells in cases of FSGS to the vascular smooth muscle-like cells that are one of the principal cell types found in ATH plaques [32]. Additionally, both diseases exhibited monocyte-derived foam cells, another key cell type in ATH [32]. These lipid-laden foam cells may be derived from a variety of cell types, including smooth muscle cells, mesangial cells, epithelial cells, endothelial cells, or resident phagocytes [33], but most glomerular foam cells are derived from CD68+ macrophages, which transform into foam cells by ingesting lipids within the glomerulus “in situ” [34]. In addition, foam cells are frequently found in the interstitium in nephrotic states and in Alport syndrome, and it has been suggested that they contribute to progressive tubulointerstitial injury [35]. In this work, we have used the ApoE^−/−^ mice that develop severe hyperlipidemia due to an accumulation of chylomicrons and Very-Low-Density Lipoprotein (VLDL) and lipoprotein glomerulopathy, making this a model of hyperlipidemic renal injury [19] to study the impact of sex on lesion development.

The most significant finding of our study was the detection of (i) increased mesangiolysis in conjunction with glomerular foam cell accumulation and (ii) swollen parietal epithelial cells (PECs) hypertrophy in male mice only, which suggests a sex predisposition to podocyte injury and contributes to the development of FSGS. Although glomerular foam cells have been associated with extremely high levels of serum lipids, there are also glomerular foam cells not associated with hyperlipidemia in patients with FSGS [36]. These findings suggested that additional factors account for the formation of glomerular foam cells. In support of this hypothesis, a transgenic murine model of FSGS demonstrated that an initial podocyte injury promoted lipid deposition and specific peroxidation that further produced a particular glomerular microenvironment, facilitating macrophage recruitment and promoting foam cell formation at the site of injury [37]. In addition, dissolution of the normally compact mesangial matrix (“mesangiolysis”) is frequently present with glomerular foam cell accumulation, but it remains unknown whether these processes are mechanistically linked.

To date, there has been little evidence to link glomerular foam cell accumulation to podocyte injury [32]. Currently, PECs are receiving great attention as progenitor cells of podocytes and as barrier-forming cells that prevent leaks in the periglomerular ultrafiltrate. PECs contribute to the thickening of the Bowman’s capsule basement membrane through the secretion of collagen, which has been observed in diabetic nephropathy [38]. PECs also may transdifferentiate into podocytes in response to severe glomerular injury [31]. In this sense, it has been reported that hyperglycemia induces cellular hypertrophy in podocytes, mesangial cells, and renal tubular cells [39]. ACEi treatment demonstrated a reduction in PECs in animal models of human immunodeficiency-associated nephropathy (Tg26 mice) [40] and enhanced the ability of PECs to turn into progenitor cells in in a model of progressive glomerular lesions in rats [41]. In our study, we detected a regression in PEC hypertrophy but not in proteinuria.

FSGF has a male sex predominance, and it is known that kidneys from males and females show fundamental differences in their structures [42]. Thus, the cortical proximal tubule volume density and cell height in females were 75% of that in male kidneys, but, nevertheless, female kidneys presented a higher proportion of collecting duct volume density and larger intercalated cells when compared with male mice [43]. Furthermore, these decreased total kidney size, proximal tubule density, and proximal tubule height were reversed with testosterone replacement in a C57BL/6 mice model of orchiectomy [36]. In addition, it has been shown that the androgen receptor (AR) is expressed in both male and female kidneys, but only males showed the AR in glomerular PECs [44]. Thus, in our study, we observed that under stimulation only male mice present changes in the structure of the parietal lamina of Bowman’s Capsule that were associated with more proteinuria and poorer renal function.

The application of WSI is aimed at overcoming poor intra-observer and inter-observer reproducibility of scoring systems [45]. The NEPTUNE Pathology Scoring System (NPSS) provides a comprehensive scoring system by utilizing “descriptors” for patterns of glomerular and podocyte injury in each glomerulus rather than assigning a conventional diagnosis to each biopsy [46]. NPSS with validated descriptors has been proposed as a model for the standardization of renal biopsy interpretation [17]. In this work, we used a well-established DeepLab V3 model based on a semantic segmentation CNN-based architecture with a pre-trained ResNet-18 backbone based on the Microscope Image Browser (MIB2), a high-performance MATLAB-based software for the automatic identification and segmentation of glomeruli within kidney Whole Slide Imaging [47]. The DeepLab architecture employs the atrous convolutions that allow tuning the right balance between context assimilation (large field of view) and fine localization (small field of view), avoiding the loss of spatial information [48]. Our results showed a mesangial expansion due to glomerular foam cell accumulation with a clear reduction in the urinary space in male mice in both the PBS group and the enalapril-treated group when compared to female mice. These results were validated by two independent observers and obtained good performance in the segmentation task with good prediction precision. Thus, we suggest that semiautomatic or full segmentation methods can help achieve a rapid and accurate characterization of glomerular lesions.

## 4. Materials and Methods

### 4.1. Reagents

In this work we used the following reagents from the stated suppliers: enalapril (E6888, Sigma-Merck, St Louis, MO, USA), Angiotensin II (A9525; Sigma-Merck, St Louis, MO, USA), isoflurane (PDG96236, Baxter Corporation, Deerfield, IL, USA), Hematoxilin (HHS32-1L; Sigma-Merck, St. Louis, MO, USA), Oil-Red-O (O0625; Sigma-Merck, St. Louis, MO, USA) DAB (MKCK2487, Sigma-Merck, St. Louis, MO, USA), CV Mount mounting medium (14046430011, Leica Biosystems, Wetzlar, Germany), Masson trichromacy (1004850001; Sigma-Merck, St Louis, MO, USA), F4/80 (HM1066 Hycult Biotech, Uden, The Netherlands), anti-NF-κB (ab7970 Abcam, Cambridge, UK), Goat polyclonal antibody Vectastain ABC Kit (PK-4000, Vector laboratories, Los Angeles, CA, USA), and IgG2a goat anti-rat (NB7122, Novus Biologicals, Centennial, CO, USA) C57BL/6J ApoE^−/−^ mice (Jackson Laboratories, Bar Harbor, ME, USA).

### 4.2. Mice and Experimental Groups

Experiments were performed in 8-week-old ApoE^−/−^ mice on C57BL/6 background (12 males and 12 females; Jackson Laboratories, Bar Harbor, ME, USA), fed for 10 weeks with a high-fat rodent diet that contained 1.25% cholesterol and provided 40% of the energy as fat (D12108CI; Research Diets Inc., New Brunswick, NJ, USA). The design of the study is summarized in Figure 4. Mice were randomized into three different groups: (i) an enalapril group (n = 12; 5 mg/kg/day enalapril dissolved in PBS and administered subcutaneously, (ii) a group treated with Ang II (n = 6; 6 μg/kg/min Ang II for 2 weeks) delivered through a subcutaneously implanted osmotic minipump (Model 1002 Micro-Osmotic Pump Alzet, Durect Corporation, Cupertino, CA, USA) with a pumping rate of 0.25 uL/h in order to identify renal lesions caused by Ang2, and (iii) a control group treated with the vehicle PBS for an additional period of 4 weeks (n = 12). Animals were euthanized at 22 weeks of age by inhalation of <5% isoflurane and cardiac puncture. Next, the vascular tree was perfused with PBS under physiological pressures, and aortas and kidneys were removed, and fixed with 4% paraformaldehyde during 12 h for aortas or with 10% buffered formalin during 2 days for kidneys. All animal studies were carried out in accordance with recommendations in the Guide for the care and use of Laboratory Animals of the National Institutes of Health. The protocol was approved by the Committee on the Ethics of Animal Experiments of UB-Bellvitge (number 85/20).

### 4.3. Blood Pressure Analysis

BP was non-invasively measured by using a tail-cuff sphygmomanometer (BP-2000 Blood Pressure Analysis System, NIBP LE5001, PANLAB) on conscious, restrained mice, before the initiation of the treatment and at the end of it. In order to reduce spontaneous variation, mice were trained for a period of 2 days to become used to the tail-cuff method. Prior to BP readings, mice were optimally warmed. All BP measures were calculated as the average of 5 measures per day.

### 4.4. Blood and Urine Analysis

Total cholesterol, HDL, and TG measurements were performed on plasma at the end of the study. Every 24 h urine samples were collected in metabolic cages at weeks 18 and 22 of the study. Plasma creatinine and urinary concentration of protein were measured at the Clinical Veterinary Biochemistry Services of the University of Barcelona. Animals were allowed free access to water and food.

### 4.5. Atherosclerotic Lesions Analysis

At the time of sacrifice, aortas were removed, dissected, and fixed in 4% paraformaldehyde in PBS for 12 h. The “en-face” area of the aortic arch and descendent aorta were cut open, pinned out flat on a black surface, and stained with Oil-Red-O (ORO). The percentage of the aortic area covered with lipid-containing plaque was measured in the whole aorta. Hearts were perfused with PBS, removed by cutting the aortic root, and further fixed in 4% paraformaldehyde in cold PBS for 12 h. These were further drained and rinsed overnight in PBS containing 20% sucrose. Hearts, aortic roots, and arches were embedded in OCT, and tissue blocks were cut into serial 10 μm thick sections from the proximal aorta, beginning at the end of the aortic sinus [49]. The composition of the plaques in the aortic root was analyzed by staining with ORO and counterstaining with haematoxylin. Cryosections were stained either with Masson’s trichrome staining for collagen, with anti-F4/80 antibodies to determine macrophage content, and with anti-NFκΒ for inflammation. Images of each aorta were captured with a digital camera mounted on a dissection microscope. The plaque area was quantified morphometrically using the ImagePro software by measuring the extent of surface area covered by neutral lipids as revealed by ORO staining in a series of 6 sections on immediately adjacent slides that had a common anatomic location (170 μm after the appearance of the 3rd valve) at the coronary ostium by calculating the percentage of ORO staining.

### 4.6. Renal Histology Analysis

In order to evaluate the degree of injury in the kidneys, sections were stained with Masson trichrome. A total of 15 non-overlapping random field photomicrographs were acquired per sample with the Nikon Epifluorescence microscope (E800) with 40× magnification and imported into ProgRes Capture (Jenoptik, Jena, Germany). The Bowman’s capsule area (BA) was defined as the area of the inner side of the glomerular parietal epithelial cell layers, and the glomerular area (GA) was defined as the area of the outer capillary loops of the tuft [50]. The proportion of urinary space (US) was calculated as US = BA-GA/BA. The number of nuclei, BA, and GA were manually quantified for each glomerular cross-section from a total of 15 glomeruli using the ImageJ2 v 2.9.0/1.53 t software. Each glomerulus was evaluated for the presence of foam cells and for histopathologic changes in parietal epithelial cells (PECs). We discriminated between quiescent and activated PECS internally lining the Bowman’s capsule by morphological analysis. Quiescent PECs were flat, and the activated PECs had enlarged nuclei and increased cuboidal cytoplasm [51]. The numerical density of activated PECs in glomerulus (Nv(P/Glom)) was calculated by dividing the absolute PEC number per glomerulus (Q(P)) by the area of glomerular cross-sections (A(Glom)), according to the formula: Nv(P/Glom) = (Q(P)/ h × A(Glom)) × fs3, where h = 1.5 was the dissector height and fs3 = 0.95 the linear tissue shrinkage correction factor (for Epon-embedded murine kidney tissue) [52]. Interlobular artery injury was determined by analyzing wall thickness after tracing the inner and outer circumferences of the vessel. The size of the vessel was normalized by calculating the ratio of median wall area dimension to the area of outer circumference. All the measures were obtained blindly by two investigators. Alternatively, for automatic and reliable identification of glomerular area and urinary space, we obtained a whole slide image (WSI), from Masson’s trichrome stained samples using a 3d Histech P250 scanner, at 40× magnification and transformed into monofocal by using Extended Depth of Focus. A total number of 141 glomeruli were selected and 4 categories were defined manually as “annotations” (urinary space, glomeruli, tuff area, and exterior), and a mask was extracted by using the QuPath (QUP) v.0.4.2 software. Each glomerulus was identified and delimitated with tiles of 349 × 349 pixels and manual segmentation of Urinary space was performed using the “wand” tool. Masks and original images were split into the raining group (n = 91) and validation group (n = 50) and processed using a deep learning semantic segmentation algorithm that classifies every pixel in an image following the code-free approach proposed by Pettersen et al. in the microscopy image segmentation software DeepMIB2 [53]. Briefly, tile images and masks were exported in .tif and .png file formats, respectively, to train the DeepLabV3Resnet18 network, a convolutional network (CNN) for semantic segmentation, using the DeepMIB2 v 1.302.0.0 (Microscopy Image Browser 2, licensed under GNU General Public License v2, and developed during 2010–2021 by Electron Microscopy Unit of Institute of Biotechnology, University of Helsinki, Finland) software (http://mib.helsinki.fi/; accessed on 21 February 2023). The training step encompassed 100 epochs, 4 iterations, a total of 141 images (adding data augmentation such as flips and rotations among others), using 0.25 fraction of validation and 16 mini-Batch size.

Kidney sections were also stained with anti-F4/80 (1/50) to study the number of interstitial macrophages and with an anti-NF-κB (1/1000) in order to evaluate inflammation in the kidney. Appropriate secondary antibodies IgG2a goat anti-rat (1/100) and Goat polyclonal antibody Vectastain ABC Kit (1/200) were used before DAB development. Finally, sections were counterstained with Harris hematoxylin and covered with CV Mount mounting medium. F4/80 or NF-κB-positive cells were quantified in 43 glomeruli. Fluorescence intensity was graded by using a semiquantitative scale from 0 (negative) to 4 (very strong). For all histological analyses, simultaneous negative control staining reactions were performed without the primary antibody. The results represent the percentage of glomeruli that contained positive cells.

### 4.7. Statistical Analysis

Data were expressed as mean ± SD. Blood pressure changes, biochemistry parameters, and the extension of atherosclerosis and renal lesions were determined using the nonparametric Mann–Whitney or Kruskal–Wallis test. The level of significance was *p* < 0.05. Statistics were performed with R version 4.1.3 (10 March 2022).

## 5. Conclusions

In conclusion, in this work we show that treatment with enalapril reversed the plaque burden in the whole aorta of male animals but did not have a significant effect on the plaque size measured in the aortic root. Additionally, in the ApoE model of ATH, male mice exhibited a more severe glomerular injury compared to females. This was evident from an increased accumulation of glomerular foam cells and activation of parietal epithelial cells (PECs), both suggesting a functional link among lipid metabolism, podocyte injury, and sex hormones. Treatment with enalapril offered renal protection, particularly in females as well as the normalization of PEC function. Thus, the optimal treatment of ATH will require the development of a personalized and integrative strategy that combines therapies against established risk factors, such as hypercholesterolemia or hypertension, with novel drugs targeting altered key molecular mechanisms identified in ATH progression.

## Figures and Tables

**Figure 1 ijms-24-13442-f001:**
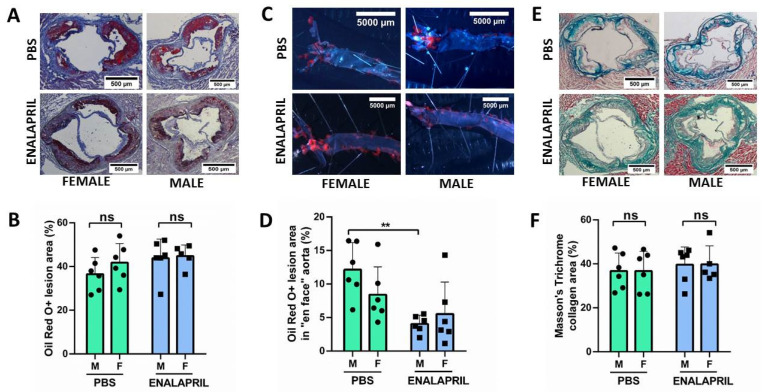
Treatment with enalapril does not regress atherosclerotic lesions in ApoE^−/−^ mice. (**A**) ORO staining of a representative aortic sinus of 22-week-old ApoE^−/−^ mice, female (left) or male (right), after 4 weeks of enalapril, 5 mg/kg/day, or PBS. Scale bar = 500 μm. Black arrows indicate ATH. (**B**) Lesion quantification was performed with ImageJ2 v 2.9.0/1.53t using the sum of all the individual areas measured throughout the valve (6–8 sections). Data are represented as mean ± SEM. (**C**) Representative en face aortas stained with ORO. Scale bar = 5000 μm. Red stains show ATH plaques. (**D**) Quantification of the ORO-positive areas of the en face images. Enalapril reduced the number of plaques in ApoE^−/−^ males compared with the PBS group. (**E**) Representative aortic sinus stained with Masson’s trichrome for quantification of collagen patches (in blue). Scale bar = 500 μm. (**F**) Quantification of Collagen deposition in aortic roots was similar in all groups. Kruskall Wallis test; ** *p* < 0.01. Abbreviations: ORO: Oil Red O; M: Male; F: Female; PBS: phosphate-buffered saline; ns = nonsignificant; SEM: Standard error of the mean.

**Figure 2 ijms-24-13442-f002:**
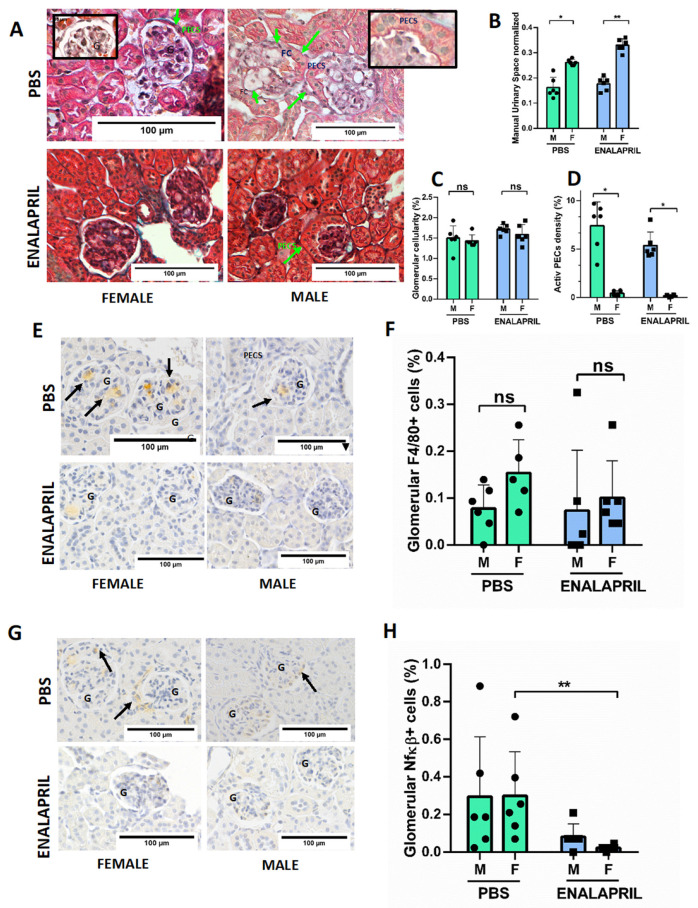
Glomerular injury was increased in male ApoE^−/−^ mice. (**A**) Representative glomerulus (**G**) from females (left) or males (right). Green arrows indicate urinary space (upper), while foam cells, and parietal epithelial cells are shown by letters (FC and PECS, respectively). The window in the right-upper panel shows activated PECs. (**B**) Quantification of the urinary space in 15 glomeruli from each mouse with ImageJ2 v 2.9.0/1.53t (n = 6 male; n = 6 female) showing its reduction in males. Scale bar = 100 μm. (**C**) Quantification of mean mesangial cellularity showed no sex-associated differences or according to the enalapril treatment. (**D**) Quantification of activated PECs in glomeruli showed their increase in males. (**E**) Representative glomerulus showing macrophages as F4/80+ cells (black arrows). (**F**) Quantification of total macrophage (F4/80+ cells) infiltration. (**G**) Representative glomerulus showing NF-κB+ cells (black arrows). (**H**) Quantification of NF-κB+ cells. Kruskal-Wallis was performed for statistical analyses; * *p* < 0.01; ** *p* < 0.0001. Abbreviations: G: Glomerulus; PECs: parietal epithelial cells; FC: foam cells, M: Male; F: female; PBS: phosphate-buffered saline; ns = nonsignificant.

**Figure 3 ijms-24-13442-f003:**
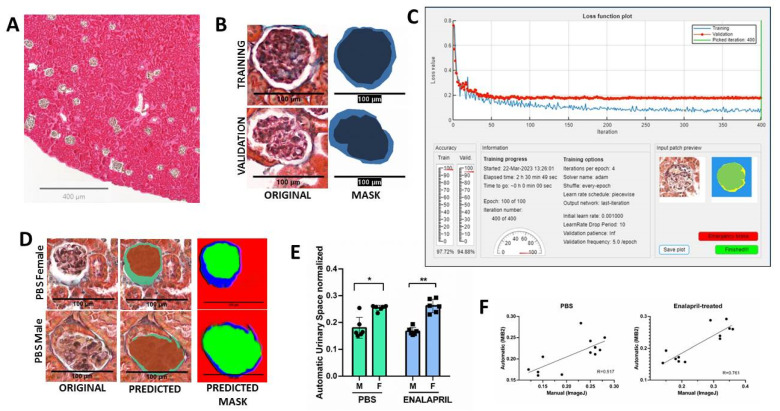
Examples of deep learning quantification of glomerular lesions. (**A**) Representative whole slide image (WSI) used for glomerular selection in QuPath (QUP) software. Scale bar = 400 μm). (**B**) A total number of 141 glomeruli were selected in QuPath (ORIGINAL) and 4 categories defined (urinary space, glomerulus, tuff area, and exterior) to extract a mask using a QUP script (MASK). (**C**) Representative loss function plot of trained DeepMIB2 software. Blue lines in the plot showed that the system was trained and stabilized around itineration 100 (*x* axis) with 97.72% of accuracy while the validation phase (red lines) was stabilized around 50 itinerations with 94.88% of accuracy. Information about the number of itinerations and training progress is also shown. (**D**) A representative glomerulus (ORIGINAL) loaded into the DeepMIB2 trained and validated network as a test-set category for class prediction for PBS (upper images) and enalapril-treated (lower images) groups; Scale bar = 100 μm. The DeepMIB2 software predicted the glomerular (brown area in the PREDICTED images) and urinary space areas (green area from PREDICTED images); Scale bar =100 μm. (**E**) Quantification of normalized urinary space that showed a reduced reduction in males. (**F**) Correlation between DL method (*y* axis) and manual quantification with ImageJ software (*x* axis). The upper graph represents PBS mice group and lower graph represents the enalapril-treated mice group. Kruskal-Wallis; * *p* < 0.01; ** *p* < 0.0001. Pearson’s correlation indexes (R) are shown. Abbreviations: M: Male; F: female; PBS: phosphate-buffered saline.

**Figure 4 ijms-24-13442-f004:**
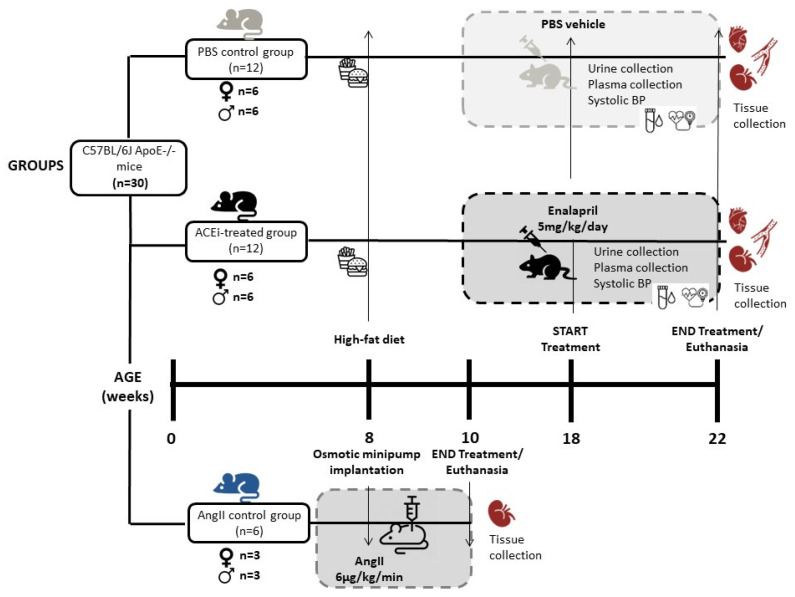
Diagram of the experimental study design, presenting the mice groups, chronogram in weeks, interventions, and sample collection.

**Table 1 ijms-24-13442-t001:** Basic mice biological characteristics.

	Control	Control		Enalapril	Enalapril	
	Female	Male	*p*	Female	Male	*p*
n	6	6		6	6	
weight (g)	21.91 (0.58)	30.91 (1.2)	0.17	24 (3)	31 (1.9)	0.5
LDL-cholesterol (mmol/L)	25.96 (8.11)	11.45 (3.96)	0.02	19.92 (10.98)	11.68 (5.17)	0.008
HDL-cholesterol (mmol/L)	2.28 (0.45)	2.38 (0.43)	0.6	2.44 (0.13)	2.64 (0.14)	0.9
TG (mg/dL)	52.28 (15.89)	84.20 (21.49)	0.5	59.8 (3)	57.7 (7.98)	0.045
Creatinine (mg/dL)	1.1 (1.09)	4.06 (2.1)	0.012	2.04 (2.53)	1.72 (1.46)	0.5
24 h Proteinuria (mg/DL)	122 (58)	1122 (498)	0.016	105 (78)	1146 (290)	0.037
TAS (mmHg)	137 (12)	107 (25)	0.2	108 (12)	101 (15)	0.7

## Data Availability

The data generated during this study are available upon request.

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
