# Peer review of "Sex Differences in Glomerular Lesions, in Atherosclerosis Progression, and in the Response to Angiotensin-Converting Enzyme Inhibitors in the ApoE^−/−^ Mice Model"

_ijms, 2023, doi:10.3390/ijms241713442_

Round 1

Reviewer 1 Report (Previous Reviewer 2)

Authors corrected paper. I have no more comments.

Author Response

Thank you for your comments.

Reviewer 2 Report (Previous Reviewer 1)

The work does not have high merit and is not innovative. However, I believe it can be published, as it contains information that supplements the commonly accepted knowledge.

Author Response

Thank you for your comments.

This manuscript is a resubmission of an earlier submission. The following is a list of the peer review reports and author responses from that submission.

Round 1

Reviewer 1 Report

Its true, that the observations of atherosclerosis progression, and the response to angiotensin-converting enzyme inhibitors in The Apoe-/- Mice model potentially has sense, because atherosclerosis and hypertension are associated with changes in structural and functional parameters of the vascular wall, but I am afraid that the treatment with of angiotensin-converting enzyme (ACE) inhibitor enalapril may be the result of plaque fissures coronary thrombosis.

Action at all stages of the atherosclerotic process can only be achieved through aggressive therapy aimed at the intensive reduction of risk factors. However, optimal treatment of atherosclerosis requires not only therapies that directly affect risk factors, such as hypercholesterolemia or hypertension, but drugs that affect the atherosclerotic process itself.

Differences in the effects of enalapril on male and female mice add little information regarding the problem of atherosclerosis, so why did the authors use the Apoe-/- Mice Model?

Author Response

Thank you to the reviewer for their positive comments. We have addressed each comments point-by-point and incorporated your feedback to improve the paper.  

It’s true, that the observations of atherosclerosis progression, and the response to angiotensin-converting enzyme inhibitors in The Apoe-/- Mice model potentially has sense, because atherosclerosis and hypertension are associated with changes in structural and functional parameters of the vascular wall, but I am afraid that the treatment with of angiotensin-converting enzyme (ACE) inhibitor enalapril may be the result of plaque fissures coronary thrombosis.

I agree with the reviewer observation that atherosclerosis and hypertension are associated with complex structure and function changes of vascular wall (Lee R et al. Hypertension Research 2017) and ACEi have significant effects on vascular structure. Thus, we have added this topic in the discussion.

Lines 216-220:

"Although, we did not observe any significant differences in the degree of inflammation or plaque size between male and female mice, treatment with enalapril reversed plaque burden in the whole aorta of male animals. However, enalapril did not have a significant effect on plaque size measured with Oil Red O in the aortic root".

Lines 225-253:

"It has been demonstrated that ATH and hypertension are associated with complex structure and function changes of vascular wall and ACEi have significant effects on vascular structure [21]. In adult (15-week-old) spontaneously hypertensive rats (SHR), treatment with quinapril for 10 weeks caused vascular remodeling by inducing apoptosis in medial smooth muscle cells (SMC) in mesenteric arteries [22]. However, the response to ACEi has been observed to differ between arteries, suggesting that the specific composition vary within and between vascular beds [21]. Furthermore, this heterogeneity observed in the response to ACEi can be influenced by various factors, such as previous vascular lesions, aging, sex or the species used in the experimental model. This diversity in the response was observed in young spontaneously hypertensive rats (SHR), where enalapril treatment prevented further development of vascular hypertrophy in the renal vessels. In contrast, in adult SHR (18 to-19-week-old) with established hypertension, long-term enalapril treatment cause a significant reduction of the media area in the interlobar arteries but not in the interlobular arteries or in the afferent arterioles [23]. In our experiment, we observed that treatment with enalapril for 4 weeks resulted in a reversal of plaque burden in the whole aorta of male animals. However, this treatment did not have a significant effect on plaque size measured with Oil Red O in the aortic root. These data suggest that other factors, such as the perivascular adipose tissue (PVAT) could contribute to this heterogeneity in the reversion of vascular lesions [24].

In addition, male animals have been reported to have more inflamed, yet smaller plaques compared to female animals [8]. However, most preclinical studies using animal ATH mice models did not examine both sexes, and even in those that do, well-powered direct statistical comparisons for sex as an independent variable remain rare [8]. It has been reported that female mice have larger plaques than their male littermates up to 6 months of age, but after 8 months old ApoE-/-, several studies have observed that males have equal or even larger plaques [8]. In our experiment, we did not observe any significant sex-related differences in plaque size measured in the aortic root at 22-week-old ApoE-/- mice fed with 14 weeks with a high fat diet (Figure 1B)".

Action at all stages of the atherosclerotic process can only be achieved through aggressive therapy aimed at the intensive reduction of risk factors. However, optimal treatment of atherosclerosis requires not only therapies that directly affect risk factors, such as hypercholesterolemia or hypertension, but drugs that affect the atherosclerotic process itself.

 I agree with this point of view and have included a comment in the conclusions section.

Differences in the effects of enalapril on male and female mice add little information regarding the problem of atherosclerosis, so why did the authors use the Apoe-/- Mice Model?

We have extended the discussion to include the results on atherosclerosis. Additionally, we have included the evaluation of plaque inflammation. The ApoE-/- model was chosen because our focus is on atherosclerosis and its relationship with kidney disease. Therefore, the ApoE-/- model mouse is a standard model for studying ATH and has also been proposed as model of renal injury.

Reviewer 2 Report

Adrián Mallén et al. submitted original paper entitled Sex Differences In Glomerular Lesions, Atherosclerosis Progression, And In The Response To Angiotensin-Converting Enzyme Inhibitors In The Apoe-/- Mice Model.

Authors analyze sex-based differences in renal structure and response to the Angiotensin-Converting Enzyme (ACE) inhibitor enalapril in a mouse model of atherosclerosis.

The study demonstrates sex-based differences in the response to enalapril in a mouse model of atherosclerosis. Males exhibited more severe glomerular injury, while enalapril provided renal protection, particularly in females. These findings suggest potential sex-specific considerations for ACE inhibitor therapy in chronic kidney disease and atherosclerosis cardiovascular disease. However, further advanced research is needed. It can be interesting information for readers and medical specialists. Manuscript is well prepared from the technical and meritorical side. Eventually, conclusions could be etended. I have no serious comments. In my opisnion, paper could be accepted in the present form.

Author Response

Thank you to the reviewer for their positive comments. We have addressed each comments point-by-point and incorporated your feedback to improve the paper.

Adrián Mallén et al. submitted original paper entitled Sex Differences In Glomerular Lesions, Atherosclerosis Progression, And In The Response To Angiotensin-Converting Enzyme Inhibitors In The Apoe-/- Mice Model.

Authors analyze sex-based differences in renal structure and response to the Angiotensin-Converting Enzyme (ACE) inhibitor enalapril in a mouse model of atherosclerosis.

The study demonstrates sex-based differences in the response to enalapril in a mouse model of atherosclerosis. Males exhibited more severe glomerular injury, while enalapril provided renal protection, particularly in females. These findings suggest potential sex-specific considerations for ACE inhibitor therapy in chronic kidney disease and atherosclerosis cardiovascular disease. However, further advanced research is needed. It can be interesting information for readers and medical specialists. Manuscript is well prepared from the technical and meritorical side. Eventually, conclusions could be extended. I have no serious comments. In my opisnion, paper could be accepted in the present form.

Thanks for your valuable comments. We have expanded the discussion on results atherosclerosis and also expanded the conclusions section.